# Differential provisioning roles, prey size, and prey abundance shape the dynamic feeding behavior of gray wolves

Thomas D. Gable [1,3✉], Sean M. Johnson-Bice[2,3], Austin T. Homkes[1] & Joseph K. Bump[1]

The demands of raising dependent young can influence the feeding behaviors of social carnivores, especially for individuals that are primarily responsible for provisioning young. We investigated how the feeding and provisioning behavior of a social carnivore, gray wolves (*Canis lupus*), are connected and shaped by extrinsic and intrinsic factors, and whether and how these patterns changed throughout the pup-rearing season (April–August). We found breeding wolves had shorter handling times of prey, lower probability of returning to kills, and greater probability of returning to homesites after kills compared to subordinate individuals. However, the feeding and provisioning behaviors of breeding individuals changed considerably over the pup-rearing season. Wolves had longer handling times and returned to provision pups directly after kills less frequently as annual prey abundance decreased. These patterns indicate that adult wolves prioritize meeting their own energetic demands over those of their pups when prey abundance decreases. We suggest that differential provisioning of offspring based on prey abundance is a behavioral mechanism by which group size adjusts to available resources via changes in neonate survival.

[1]Department of Fisheries, Wildlife and Conservation Biology, University of Minnesota, St. Paul, MN, USA. [2]Department of Biological Sciences, University of Manitoba, Winnipeg, Manitoba, Canada. [3]These authors contributed equally: Thomas D. Gable, Sean M. Johnson-Bice. ✉email: thomasd.gable@gmail.com

Understanding the factors that shape foraging behavior and parental care when raising dependent young has been a longstanding, fundamental focus in the ecology of social animals. In social carnivores, groups of breeding and non-breeding individuals help raise dependent young directly (e.g., provisioning, watching/defending) and indirectly (e.g., cooperative hunting of prey, territory maintenance)[1–3]. The dependent young of most social carnivores are kept at specific locations such as dens or burrows from which adults radiate out in search of prey or other food[4–6]. Once prey or food are acquired, many social carnivores transport food back to rearing sites to provision dependent young[7,8]. The need to meet the energetic demands of dependent young can influence the foraging (e.g., movement patterns, predation behavior) and feeding behavior (e.g., handling time, carcass attendance) of social carnivores, especially for individuals that are primarily responsible for provisioning[9,10]. Despite this, feeding and provisioning behavior in social carnivores are often examined separately and not as linked biological processes, which has limited our understanding of important behavioral strategies social carnivores use when rearing young.

Simultaneously studying and understanding the interplay between feeding and provisioning behaviors is often challenging because many social carnivores are cryptic, sensitive to human observers, and often travel large distances to procure food for dependent young. Even in environments where social carnivores are readily observed, studying how feeding and provisioning behaviors are connected is challenging given the extensive movements by many social carnivores to find prey[3,11,12]; it is hard for researchers to be in two different places—kills and rearing sites—at the same time. However, the combination of GPS-collar technology and field investigations of GPS-locations to identify predator-killed prey provides an indirect method to examine both provisioning and feeding behaviors. Indeed, with GPS-collars, researchers can estimate the handling time of prey by carnivores[13,14], determine patterns of carcass attendance and visitation[15], and identify recursive movements to and from

rearing sites[16,17]. Yet, previous work has not, to our knowledge, examined the way these various behaviors are linked, and, more importantly, how intrinsic (e.g., social status, age of dependent young) and extrinsic factors (e.g., prey abundance, prey type or size) shape the relationship between feeding and provisioning behaviors.

Gray wolves (Canis lupus) are cooperatively breeding social carnivores that live in social groups (packs) that typically consist of a breeding pair and their subordinate offspring[7,18]. Breeding females give birth to pups in spring (April-May) and pups are kept at dens for their first ~8 weeks of life before they are moved to rendezvous sites (den and rendezvous sites are collectively referred to as 'homesites'). Throughout the pup-rearing season, breeding individuals and subordinate pack members generally help to rear pups by directly provisioning pups and by guarding and caring for pups at homesites[19]. Yet, the amount of time wolves spend provisioning, caring for, and rearing pups generally depends on social status[9]. Breeding individuals generally contribute more to raising pups than subordinates presumably because they, unlike subordinates, have a direct reproductive investment in their pups[7,20]. However, there is considerable variation among subordinates in their willingness to provide alloparental care for pups and the reasons for this variability are not well understood[9,21]. Some have hypothesized that the willingness of subordinates to provision and provide care for pups is dependent on prey abundance but that hypothesis is largely untested[19,21,22].

During the pup-rearing season, wolf pack cohesion is substantially reduced and wolves often hunt and kill small prey (e.g., ungulate neonates and beavers [Castor canadensis]) by themselves during this time[23–25]. For example, in northern Minnesota, collared wolves were with other collared pack members ≤6% of the time during the summer[26]. Between foraging bouts, wolves often return to homesites to provision and care for pups as well as interact with other pack members (Fig. 1;[17,27]). Because the behavior of wolves during most of the pup-rearing season

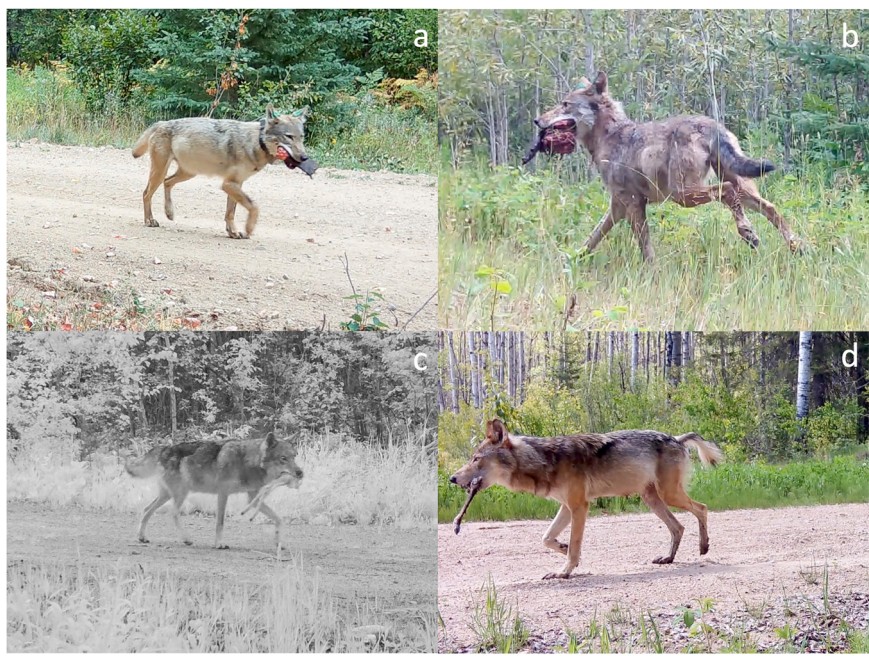

**Fig. 1 Breeding wolves transporting prey remains back to homesites to provision pups.** Examples of breeding wolves transporting the remains of beavers **a**, **b** and deer fawns **c**, **d** back to homesites to provision pups in the Greater Voyageurs Ecosystem, Minnesota. The wolves in each image have distended stomachs from consuming prey, which is not unusual because they often transport prey back to pups via their stomachs and mouths[38]. All images are stills taken from remote camera footage.

typically reflects the choices and decisions of individuals (i.e. not the larger social group)[28], wolves are an excellent model for understanding how foraging and provisioning behaviors of individuals within social carnivore groups are both connected and shaped by intrinsic and extrinsic factors.

We investigate how the feeding and provisioning behavior of gray wolves in the Greater Voyageurs Ecosystem (GVE), Minnesota, USA are shaped by social status and prey abundance throughout the pup-rearing season (April–August). Observing wolves in the dense forests of the GVE is difficult and collecting direct behavioral data not possible. We therefore used GPS-collars programmed with high fix rates, in combination with intensive ground-based investigations, to indirectly measure and assess several aspects of feeding and provisioning behavior. Collectively, our goal was to understand (1) how individual rearing roles (breeding vs. subordinate) shape the feeding behavior of a cryptic social carnivore, (2) how prey size, type, and abundance influence feeding behavior and provisioning of dependent young, and (3) how feeding and provisioning behavior change throughout the rearing season as offspring mature.

We assessed the evidence for three specific hypotheses. Our first hypothesis was that differential provisioning roles, along with the maturation of dependent young within social carnivore groups, shapes the feeding behaviors of individuals. We hypothesized that the feeding behaviors of breeding wolves would differ from subordinates because breeding individuals are disproportionately responsible for provisioning dependent young[7,20]. As such, we predicted breeding wolves would exhibit shorter prey handling times, be less likely to return to kills, and be more likely to return directly to homesites after kills than subordinate wolves. Additionally we predicted the feeding behavior (i.e., handling time, returning to kills, and returning to homesites) of breeding individuals would change as pups grow and mature, whereas the feeding behavior of subordinate individuals would remain relatively static throughout the pup-rearing season.

Our second hypothesis was that feeding behavior of social carnivores is shaped by prey size. We predicted, consistent with previous empirical and theoretical research[29–31], that wolves would have longer handling times and would return more frequently to kills of beavers than deer fawns because beavers are larger than fawns throughout most of the pup-rearing season.

Our third hypothesis was that social carnivores alter feeding and provisioning behavior in response to changes in prey abundance. Specifically, we hypothesized that as prey abundance decreases, social carnivores provision dependent young less often and instead prioritize meeting their own energetic demands before that of offspring[22]. Thus, we predicted wolves would return to homesites directly after kills less frequently as prey abundance decreased, choosing instead to return to their kills more often and spend more time at kills.

## Results

We searched 17,041 clusters of GPS locations from 32 wolves from 16 different packs during April to August from 2017 to 2022. Of those clusters searched, we identified 901 predation events (584 deer fawn kills and 317 beaver kills). We searched clusters from 13 breeding wolves and 20 subordinate wolves (one wolf was monitored for two seasons, when they went from a subordinate wolf in the first season to a breeding individual in the second season).

We examined the percent of kills collared wolves made with another collared pack member using data on 11 dyads (5 subordinate-subordinate dyads, 4 breeding-breeding dyads, and 2

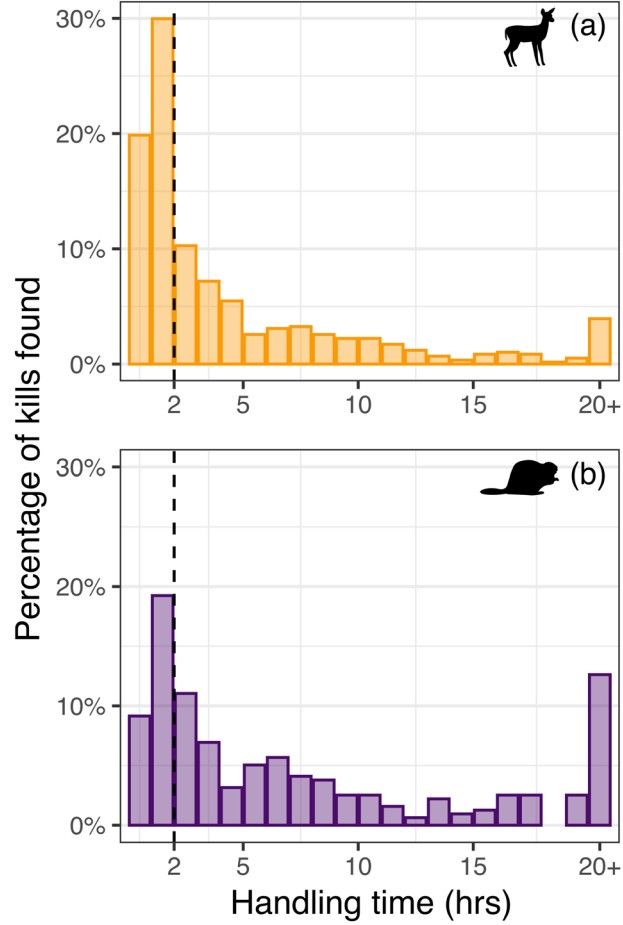

**Fig. 2 Handling times of small prey by gray wolves in the Greater Voyageurs Ecosystem, Minnesota, USA.** Handling times of white-tailed deer fawn (**a**, n = 584) and beaver (**b**, n = 317) carcasses by gray wolves in the Greater Voyageurs Ecosystem, Minnesota, USA. The dashed vertical line distinguishes between the percentage of carcasses with handling times of 0–2 h and carcasses with >2 h handling times.

subordinate-breeding dyads). Wolves in these dyads killed prey without another collared pack member at 94% of kills (312/333) during April to August.

**Influence of breeding status on prey handling times.** Median handling time of deer fawn carcasses during pup-rearing was 2.1 h and handling times for 20% (n = 116) and 50% (n = 291) of fawn carcasses were ≤ 1 hr and ≤ 2 hr, respectively (Fig. 2a). Median handling time of fawn carcasses by breeding wolves was 1.7 hr while median handling time of fawns by subordinates was 4.4 h. Median handling time of beaver carcasses during pup-rearing was 5.2 hr and handling times for 9% (n = 29) and 28% (n = 90) of beaver carcasses were ≤1 h and ≤2 h, respectively (Fig. 2b). Median handling time of beaver carcasses by breeding wolves was 4.0 h whereas median handling time of beavers by subordinates was 7.8 h. Carcass utilization averaged 99% across all fawn and beaver carcasses.

As suggested from the observed median handling times of fawn and beaver carcasses, results from our hierarchical GAM indicated handling time of fawn carcasses was significantly shorter than handling time of beaver carcasses, regardless of wolf breeding status ($\beta_{Fawn} = -0.58$, standard error [SE] = 0.08, t = −7.27, p < 0.0001). Breeding wolves had significantly shorter handling times on average than subordinate wolves, regardless of

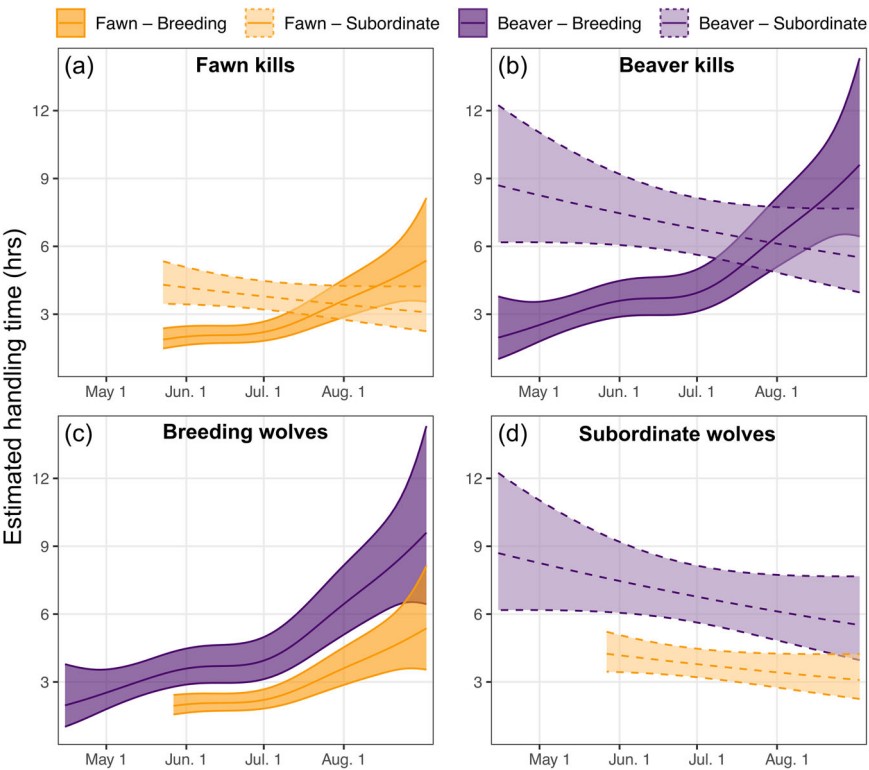

**Fig. 3 Estimated handling times of white-tailed deer fawn and beaver carcasses during the pup rearing season (April to August) for breeding and subordinate gray wolves in the Greater Voyageurs Ecosystem, Minnesota, USA. a, b** show the difference in handling time of prey between breeding and subordinate wolves while **c, d** show the same handling time estimates with a focus on comparing differences by prey type. Handling times were predicted from a hierarchical generalized additive model, with handling times of deer fawns estimated during May 26–Aug 31 (average fawn parturition date in northern Minnesota is May 26) and handling times of beavers estimated Apr. 15–Aug. 31.

prey type (fawn or beaver) ($\beta_{Subordinate} = 0.45$, SE $= 0.11$, $t = 3.98$, $p < 0.0001$), though estimated handling times for breeding and subordinate wolves were similar towards the end of the pup-rearing season (Fig. 3a, b).

Handling time of prey by breeding wolves showed a non-linear response throughout the pup-rearing season (estimated degrees of freedom [EDF] = 3.74, F = 13.6, $p < 0.0001$), with handling times increasing in duration as the season progressed (Fig. 3c). We found no evidence for non-linear handling time by subordinate wolves across the summer (EDF = 1.00, F = 2.51, $p = 0.113$; Fig. 3d). We found a negative relationship between prey biomass per wolf and prey handling time ($\beta = -1.01$, SE $= 0.43$, $t = -2.37$, $p = 0.018$) with greater intra-annual variation for breeding wolves compared to subordinate wolves (Fig. 4). The 'wolf ID' random intercept term was influential (EDF = 1.37, F = 1.15, $p < 0.0001$) while the random intercept term of 'year' was not (EDF < 0.0001, F = 0.00, $p = 0.327$).

**Probability of returning to kill site or homesite after a kill.** The probability that a wolf returned to a kill increased as the pup-rearing season progressed ($\beta_{OrdinalDay} = 0.29$, 95% confidence interval [CI] = 0.12, 0.45, $z = 3.44$, $p = 0.0006$), regardless of breeding status. However, this pattern appeared largely driven by breeding wolves; we found weak evidence for an interaction between ordinal day and breeding status ($\beta_{Subordinate*OrdinalDay} = -0.21$, 95% CI = −0.50, 0.08, $z = -1.41$, $p = 0.082$), such that the probability that a wolf returned to a kill tended to increase as the pup-rearing season progressed for breeding, but not subordinate, individuals (Fig. 5a). Overall, subordinate wolves were more likely to return to a kill than breeding

wolves ($\beta_{Subordinate} = 0.52$, 95% CI = 0.17, 0.86, $z = 2.93$, $p = 0.0034$), but this difference was most pronounced during the early pup-rearing season (Fig. 5a). Wolves were more likely to return to a beaver kill compared to a fawn kill ($\beta_{Fawn} = -0.59$, 95% CI = −0.88, −0.30, $z = -4.03$, $p < 0.0001$; Fig. 5c). Prey biomass per wolf had no influence on the probability that wolves returned to kills ($\beta = -0.62$, 95% CI = −1.98, 0.73, $z = -0.90$, $p = 0.37$; Fig. 5b). The random intercept term of 'wolf ID' was influential ($\sigma_{WolfID} = 0.174$).

We were able to determine whether or not wolves returned to homesites from 786 of the 901 kills identified in this study. For the other 115 kills, the wolf was either not using a home site or we were unable to locate one during the period when the kill occurred. We determined that the probability wolves returned to homesites following kills decreased as the pup-rearing season progressed ($\beta = -0.35$, 95% CI = −0.55, −0.15, $z = -3.40$, $p = 0.0007$). However, this pattern was largely driven by the decreasing probability that breeding wolves, in particular, returned to homesites as pup-rearing progressed; we found a significant interaction between breeding status and ordinal day ($\beta_{Subordinate*OrdinalDay} = 0.46$, 95% CI = 0.08, 0.84, $z = 2.37$, $p = 0.018$; Fig. 5d). Breeding wolves were far more likely to return to homesites after kills than subordinate wolves ($\beta_{Subordinate} = -1.36$, 95% CI = −1.94, −0.77, $z = -4.56$, $p < 0.0001$; Fig. 5d). We found a positive relationship between the probability that wolves returned to homesites and prey biomass per wolf ($\beta = 3.04$, 95% CI = 0.96, 5.12, $z = 2.86$, $p = 0.0042$; Fig. 5e). The probability that wolves returned to homesites did not differ between beaver and fawn kills ($\beta_{Fawn} = 0.17$, 95% CI = −0.18, 0.52, $z = 0.97$, $p = 0.33$; Fig. 5f). The random intercept term of 'wolf ID' was influential ($\sigma_{WolfID} = 0.561$).

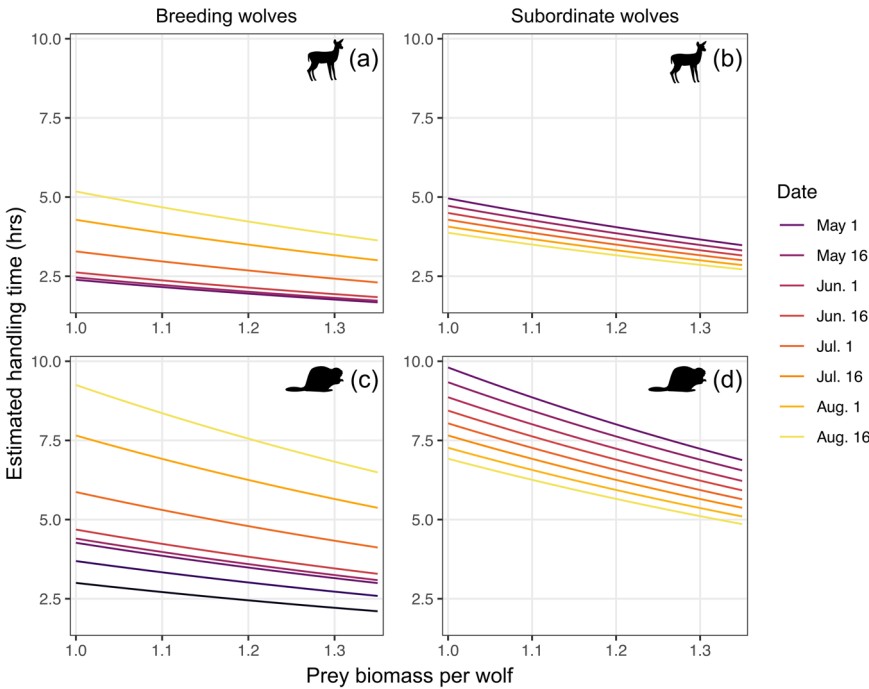

**Fig. 4 Relationship between prey biomass per wolf and estimated handling time (in hours) of white-tailed deer fawn and beaver carcasses by breeding and subordinate gray wolves in the Greater Voyageurs Ecosystem, Minnesota, USA.** Panels (**a**) and (**b**) show predicted handling times of fawn carcasses, while panels (**c**) and (**d**) show handling times of beaver carcasses. Each colored line represents the estimated relationship between handling time and prey biomass per wolf at different days during the pup-rearing season based on a hierarchical generalized additive model. Here, prey biomass per wolf was calculated by dividing the annual prey biomass index by the average annual pack size in our study area.

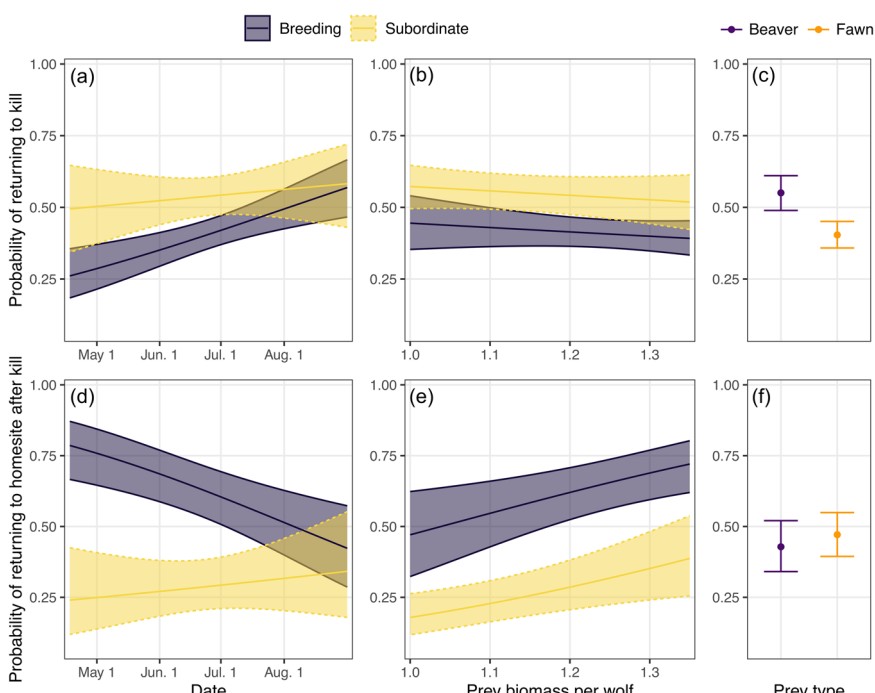

**Fig. 5 The predicted probabilities—based on breeding status and prey type—that gray wolves returned to kill sites and to homesites following kills in the Greater Voyageurs Ecosystem, Minnesota, USA.** Panels **a**, **b** show the probability of breeding or subordinate wolves returning to kills throughout the pup rearing season (**a**) and in relation to prey biomass available per wolf (**b**). **c** depicts the estimated relative probability that wolves return to beaver or white-tailed deer fawn kills. Panels **d**, **e** show the probability that breeding or subordinate wolves return to homesites directly after kills throughout the pup rearing season (**d**) and in relation to prey biomass available per wolf (**e**). **f** depicts the estimated relative probability that wolves return to homesites following beaver or fawn kills.

## Discussion

The birth of wolf pups in early April triggers a substantial change in wolf pack cohesion in the Greater Voyageurs Ecosystem as wolves transition from cooperative hunters that kill large ungulate prey (adult-sized deer) to largely solitary predators that primarily hunt and kill small prey (beavers and deer fawns). As such, studying both feeding and provisioning behavior of individual wolves provides important insight into how ecological conditions, especially the cooperative rearing of dependent young, can shape the feeding behavior of social carnivores. We demonstrate, as hypothesized, that the feeding behavior of social carnivores can be a dynamic seasonal process that is dependent on changing extrinsic and intrinsic factors. In particular, we show that feeding behavior and provisioning of dependent young differ markedly between breeding and subordinate individuals, a pattern that is likely modulated by the seasonal abundance and availability of prey, as well as the increased mobility and energetic demands of dependent young.

As predicted, handling time of prey by wolves was dependent on prey species and size. Both breeding and subordinate individuals spent considerably more time handling beavers than deer fawns (Fig. 3), likely because beavers (avg. ~15.5 kg;[32]) are substantially larger than deer fawns for most of the pup-rearing season. Most deer fawns are killed by wolves in the first 6–8 weeks of life (late May-early July) when they typically weigh 3–10 kg[33–35]. Larger prey require more time for predators to handle due to biological constraints of consumption, satiation, and digestibility[36], which is often dictated by the ratio of prey to predator body size[30]. As prey body size increases relative to predator body size, handling time increases. Thus, unsurprisingly, median handling time of beavers, which are ~55% the body mass of wolves in the GVE (28 kg;[32]), was 150% longer than median handling time of fawns. We expected handling time of fawns to increase during the pup-rearing season as deer fawns grew and increased in size, a pattern which we observed for breeding individuals but not for subordinates. Subordinate wolves rarely killed deer fawns in late summer (14 fawns killed in August by subordinates over 6 years), and we suspect we did not detect a change in handling time due to a small sample size with considerable variability.

Breeding status and the need to provision dependent young appear to be major drivers of handling time and feeding behavior in social carnivores. In many social carnivores, breeding individuals disproportionately provision dependent young, which can, in turn, alter time budgets[6,21], movement patterns[9,10], cohesion with conspecifics[25,37], and foraging behaviors[7,20] of breeding individuals. We found breeding wolves had shorter handling times, lower probability of returning to kills, and greater probability of returning to homesites after kills compared to subordinate individuals. These differential feeding patterns indicate breeding and subordinate individuals have unequal provisioning roles[7], with breeding individuals being predominantly responsible for provisioning dependent young[20,38]. Wolf pups are incapable of traveling with adults to kills during the first few months of life and instead remain at homesites[18,39]. During their first 4-5 weeks, pups are provisioned exclusively via lactation by the breeding female, who largely remains with the pups and is herself provisioned by the breeding male during this period[7,39]. Once pups are weaned, wolves must either transport food back to pups via consumption and regurgitation[38,40] or by carrying prey remains in their mouths[38] (Fig. 1), the latter of which reduces required handling time and increases provisioning efficiency. Subordinate wolves, who have reduced provisioning demands, instead returned to their kills more frequently to feed as needed. The feeding patterns of subordinates remained similar throughout the pup-rearing season while breeding individuals showed dramatic

changes, likely due to pups becoming more mature and mobile as the pup-rearing season progresses[41,42]. By 4-5 months of age, pups are capable of traveling to kills to feed directly[43] and homesites, although still used, become less important for provisioning. Thus, the demand for breeding individuals to return to homesites after kills wanes, and they can instead return to feed at the kills more frequently as the pup rearing season progresses— both of which appear to result in longer handling times.

As predicted, handling time of prey by wolves increased as annual prey biomass per wolf decreased and was accompanied by a corresponding decrease in the probability that wolves returned to homesites after kills. In other words, as annual food abundance per wolf decreased, wolves remained at kills for longer periods and returned to provision pups directly after kills less frequently—a pattern likely driven by decreased acquisition rates of prey and increased hunger by adult wolves[22]. When adults return to homesites they are mobbed by pups that vigorously lick at their mouths for food[38,41], which triggers an almost involuntary regurgitative response in adults[44]. As a result, there is likely an innate energetic cost of returning to homesites directly after kills because adult wolves might lose much of the food they recently acquired to pups[38]. To mitigate this cost when prey are less abundant, adult wolves appear to reduce how frequently they return to homesites directly after kills and instead prioritize meeting their own energetic demands over those of their pups. This does not necessarily mean that patterns in homesite attendance change when prey are less abundant—though this could be the case[21,22]. Rather, adult wolves may simply delay returning to homesites after kills to allow more time to consume and digest recently-acquired prey and ensure it is not lost via regurgitation to pups.

We think that differential provisioning of pups based on prey abundance is a behavioral mechanism by which group size adjusts to available resources via changes in offspring survival. Prey abundance and availability are major drivers of pup survival and recruitment[45–47], and, in turn, wolf pack size at localized scales[47–49]. Yet, how pack size adjusts to prey abundance is not well understood. Our work indicates that wolves provision pups less—likely by decreasing provisioning frequency and/or decreasing food delivered per provisioning bout—as prey per wolf decreases, presumably due to decreased prey acquisition, as demonstrated and expected based on the functional response of wolves[50–52]. The decrease in provisioning would undoubtedly increase pup mortality due to starvation. Such a mechanism is likely advantageous from an evolutionary perspective because it allows group size to quickly adjust to available resources via changes in pup survival so that pack and population size, which are predominantly driven by prey abundance[53,54], remain at or near carrying capacities.

Starvation is one of, if not the, most common causes of death for pups[46,55,56], which stands in stark contrast to adult wolves for which starvation is a relatively minor source of mortality[57–59]. Starvation of pups is a reflection of the foraging success of the adults they depend upon as well as the likelihood of adults to return to provision pups[22]. During periods of reduced prey abundance in northern Minnesota, wolf pups experienced substantially higher mortality rates—due primarily to starvation— than when prey were more abundant, which resulted in substantial reductions in pack size[45]. However, mortality from starvation was observed almost exclusively in pups during this period (i.e., not adults). Thus, our work, in combination with previous findings, indicates that adult wolves prioritize satisfying their own energetic demands over those of their pups during periods of reduced prey abundance.

Our detailed analysis on wolf feeding behavior during summer sheds light on the difficulties of studying and quantifying predation on small prey, particularly for cryptic predators that

cannot be directly observed[60,61]. Researchers (including this study) rely on clusters of GPS locations to locate kill sites of cryptic predators, but this can be problematic for identifying kills of small prey if cluster parameters are not sensitive enough[62–64]. Specifically, the probability of detecting kills of small prey with correspondingly short handling times will be lower if cluster parameters are too coarse[65–67]. For instance, if our GPS collars were programmed with a 1- or 2-h fix interval instead of a 20-min fix interval (as is frequently done in predation studies, e.g., Oliveira et al. [15]) we would have missed a large proportion of small prey kills because nearly one third of all beaver and half of all fawn kills had handling times ≤ 2 h. Similarly, if we defined GPS clusters based on a longer minimum time threshold spent at a kill (e.g., 2 h), we would miss a large proportion of kills even if the collar was programmed with a high fix-interval—a finding similar to several other studies[66,68–70]. Thus, it is important to have both high fix-intervals and short time thresholds for cluster definitions to adequately study predation on small prey[62]. Exact cluster parameters needed to detect predator-killed small prey may vary depending on the feeding behavior of the predator species, as coarser cluster resolutions may adequately detect kills of small prey by solitary predators such as felids[60]. Nonetheless, our work clearly shows that handling time of small prey by predators is dependent on several ecological factors and coarser cluster parameters would bias results in obvious ways (e.g., based on breeding status, prey type/size, time of year)[13,66,71]. This point is particularly relevant when indirectly estimating kill and predation rates of predators via GPS-cluster data because accurate estimates of both these metrics will depend, in part, on cluster parameters of sufficient resolution[72,73].

Studying how social carnivores cooperatively raise dependent young in dynamic and changing environments is a challenging endeavor. As a result, even for well-studied social carnivores such as wolves, there is relatively little information on predation behavior[17,74–76], provisioning of pups[38], and pup survival and mortality[46,56] during the rearing season. However, recent technological advances provide a unique, novel opportunity to understand the behavior of social carnivores and how their behavior is connected to larger predator-prey dynamics[77]. In particular, GPS-collars set to high-fix intervals to track the fine-scale movements of predators provides an effective indirect method to study foraging ecology and provisioning of dependent young[17,78]. Equally important, though substantially more challenging to obtain, are direct behavioral observations of social carnivores during this time. However, observing the behavior of cryptic social predators is becoming easier with remote video cameras and video cameras mounted on collars (not to mention opportunistic observations captured on phones[79]). Documenting and describing behavioral observations are key because they provide a lens through which indirect data and patterns can be interpreted[77]. For instance, we could not directly observe provisioning of pups in our system given the densely-forested environment. However, using scrupulously recorded direct observations of pup-provisioning behavior in other systems[7,38,39,41], we can interpret patterns in wolf movements and feeding behavior to reasonably infer provisioning behavior from movement and kill site data. Thus, natural history observations—which are often thought of as trivial, not novel, and viewed derisively by some ecologists[80]—have considerable value for understanding the behavior of cryptic social predators and interpreting data collected using indirect methods.

## Methods

**Study area**. Our study was conducted as part of the Voyageurs Wolf Project, which studies the ecology of wolves and their prey in and around Voyageurs National Park, Minnesota, USA, an area we refer to as the Greater Voyageurs Ecosystem (GVE). The GVE is typical of a southern boreal ecosystem situated in the Laurentian Mixed Forest Province. The landscape is typified by dense forests (deciduous, coniferous, and mixed) and abundant lakes, bogs, and wetlands interspersed with outcrops and rocky ridges from glacial activity. The GVE has sustained dense wolf (average density of 58 wolves/1000 km²;[81]) and beaver populations (>0.47–1.0 colonies/km²;[32,82]) for >30 years. Beavers are important seasonal prey for wolves in the GVE with beaver constituting up to 42% of wolf pack diets from April to October (the ice-free season) when beavers are vulnerable to predation. White-tailed deer (*Odocoileus virginianus*) densities in the GVE generally range between 2-4 deer/km² 2 and 4 deer/km² [83]. Deer are the primary annual prey for wolves in the area with deer fawns being one of the primary prey of wolves during the summer[35]. Fawns are generally born in late-May and predation on fawns is highest until early-to-mid July, after which there is a precipitous decline in predation on deer fawns[34,84]. For more information on the GVE, see Gable et al. [85].

**Wolf capture and collaring, and estimating pack size**. During 2017 to 2022, we captured wolves using rubber-padded foothold traps (EZ Grip #7 padded traps from Livestock Protection Company) and cable restraints to fit them with GPS collars (Vertex Plus collars from Vectronic-Aerospace) programmed to take locations every 20 min[78]. All capture and handling of wolves was approved by the Institutional Animal Care and Use Committees for the National Park Service (MWR_VOYA_-WINDELS_WOLF) and University of Minnesota (1905-37051A). For additional details on capture and handling of wolves, see Gable et al. [85].

We searched clusters of GPS-locations to locate kill sites during the pup-rearing season (April to August). We considered a cluster to be ≥2 consecutive locations ≥20 min apart and within a 200 m radius of one another[78,86]. We assumed any cluster <500 m from any occupied wolf homesite (den or rendezvous site) was associated with the homesite and not related to predation events[11,21]. We systematically searched areas around cluster locations to identify kills (for additional details *see* Gable et al. [78]). Clusters at kill sites were searched within 5.3 days, on average, after kills occurred. Once kills were found, we recorded the species killed and estimated carcass utilization[63].

We estimated handling time of prey by calculating the amount of time a wolf spent at a kill within 5 days from the first location of the kill cluster (similar to methods used by Knopff et al. [87]). We considered a wolf to be at a kill when locations were ≤ 200 m from the kill. We considered a wolf to have returned to a kill if the wolf moved > 200 m from a kill and then returned to ≤ 200 m of the kill ≤ 5 days of the first location at the kill. Because GPS-locations yield a range of time spent at kills, we considered handling time to be the average of the minimum and maximum time spent at kills[63]. In instances where wolves returned to kills, we calculated overall handling time by adding up the minimum and maximum times spent during each visit to the kill to yield overall minimum/maximum estimates, which we averaged to estimate handling time. Notably, wolves, at times, use ambush strategies to hunt and kill beavers[63]. However, determining whether a wolf used ambush strategies to kill a beaver—and if so, how much time a wolf waited-in-ambush—is not currently possible with GPS-collar data (see Gable et al. [78]. for a detailed discussion). Thus, given these limitations, our estimates of handling time would include any time a wolf spent waiting-in-ambush for beavers prior to making a kill.

We recorded whether wolves returned to homesites directly after kills based on wolf movements. Specifically, we assumed that

wolves returned to homesites directly after a kill if wolves went from a kill to an active homesite without stopping long enough to form a cluster of GPS-cluster locations[14,17]. We assumed that this direct movement from a kill to a homesite—where there are dependent pups—was a proxy for provisioning behavior in wolves, which is consistent with direct observations of wolf provisioning behavior[7,20,39,41,88]. Wolves provision pups directly or indirectly (e.g., provisioning lactating females) throughout most of the pup-rearing season by transporting kills back to homesites via their mouths or stomachs (Fig. 1;[38,39,88]). Therefore, the direct movement between kills and homesites is an indirect but logical way to assess the willingness of individual wolves to provision dependent young.

We estimated how frequently wolves cooperatively killed prey with pack members during the pup-rearing season using data from packs where we studied the predation behavior of ≥ 2 wolves at the same time[75]. We considered wolves to have cooperatively killed prey when both wolves were first present at a kill site (<200 m from kill) at the same time based on GPS-locations.

**Estimating prey abundance**. We estimated the density of deer and beavers annually during 2017 to 2022. Deer density was estimated using pellet count surveys where we counted the number of deer pellets along transects across the GVE. We used the same method and protocol described in detail in Gable et al. [35]. We estimated beaver density using aerial fall surveys where we identified and mapped all active beaver lodges in Voyageurs National Park based on the presence of a food cache, fresh cuttings, or recently maintained dam or lodge (see Gable et al. [85]. for more details on survey method). Beavers commonly maintain 2 or more active lodges within their primary pond or in secondary ponds adjacent to their primary pond, however only 1 lodge typically has a food cache[89]. To avoid overestimating the number of active lodges in these instances, we considered a distinct beaver lodge/colony to be any lodge with a food cache, or when no food cache was present but multiple lodges were being maintained, we considered the pond complex to represent 1 lodge or colony. We calculated density by taking the total number of distinct lodges identified in a year and dividing by the area of Voyageurs National Park (VNP) minus the area of the park's 4 large lakes— Rainy, Namakan, Kabetogama, and Sand Point—which represent large areas of open water not habitable to beavers or wolves.

We calculated a prey biomass index for each year using deer and beaver density estimates following the same method and ungulate biomass values as described by Fuller et al. [56]. Via this method, a single deer has a biomass index value of 1 and the biomass index value of all other prey are relative to the weight of a deer (75 kg). For example, a moose is a value of 6 because it is roughly 6 times the size of an adult deer and a beaver is 0.2 because the average beaver is 80% smaller than an adult deer[32]. To calculate biomass index values from deer densities, we simply multiplied deer density in a given year by 1 which yields relative deer biomass/km$^2$. We did not include moose in our prey biomass calculations because moose occur at very low densities (<0.05 moose/km$^2$;[90]) throughout much of the GVE, and are not a prey species for wolves in the GVE except on extremely rare occasions[35,84,91].

To calculate a biomass index for beavers from active beaver colony densities, we multiplied average beaver colony size in the GVE (5.3 beavers/colony[32];) by the average beaver lodge density in a given year. Doing so yielded the average number of beavers per square kilometer in that year. We then calculated a beaver biomass index value (beaver biomass per km$^2$) for each year by multiplying average number of beavers/km$^2$ by 0.2. We summed

the biomass values of beaver and deer for a given year to calculate annual prey biomass index values, which represent relative prey biomass/km$^2$. We then divided the relative prey biomass index for a given year by the average pack size for that year to estimate the amount of relative prey biomass available per wolf[51,92]. We used annual pack size data collected in the GVE from 2017 to 2022[81].

## Statistics and reproducibility

*Sample size*. We searched 17,041 clusters of GPS locations from 32 wolves from 16 different packs during April to August from 2017 to 2022. Of those clusters searched, we identified 901 predation events (584 deer fawn kills and 317 beaver kills). We searched clusters from 13 breeding wolves and 20 subordinate wolves (one wolf was monitored for two seasons, when they went from a subordinate wolf in the first season to a breeding individual in the second season). We examined the percent of kills collared wolves made with another collared pack member using data on 11 dyads (5 subordinate-subordinate dyads, 4 breeding-breeding dyads, and 2 subordinate-breeding dyads).

*Handling time analysis*. We used a hierarchical generalized additive model (GAM) with a Gaussian distribution to evaluate wolf handling time of beaver and deer fawn carcasses, the two primary prey of wolves in the GVE during pup-rearing[35,84]. The hierarchical GAM was implemented using the 'gam' function from the *mgcv* R package[93] with the following model formulation:

$$\ln(HandlingT)_{ijk} = f_{Status}(OrdinalDay_{ijk}) + Status_{ijk} + Prey_{ijk}$$
$$+ Biomass_{ijk} + Wolf_i + Year_j$$

where $\ln(HandlingT)_{ijk}$ is the natural log of the estimated handling time (in hrs) for the $k$th kill site from wolf $i$ in year $j$, and *Wolf* and *Year* are random intercept terms that are assumed to be normally distributed with a mean of 0. $OrdinalDay_{ijk}$ is the ordinal day of the $k$th kill site from wolf $i$ in year $j$, which was fit with a smoothing component $f_{Status}$ using thin plate regression splines comprised of 9 basis functions. The smoothing component $f_{Status}$ varied by the breeding status of wolf $i$ (breeding or subordinate), with individual penalties for each rank status (i.e., no global smoothing component, which is why $Status_{ijk}$ was also included as a main effect term)[94]. $Prey_{ijk}$ was a categorical variable (beaver or fawn) for prey type of the $k$th kill (beaver or deer fawn), while $Biomass_{ijk}$ was estimated prey biomass available per wolf for wolf $i$ in year $j$ to account for the fact that handling time may vary in response to prey abundance[50].

*Probability of returning to kills and homesites*. We used generalized linear mixed models with a logit link (i.e., logistic regression) to evaluate the probability that a wolf (1) returned to their kill and (2) returned to their homesite following a kill. We used the 'glmer' function from the *lme4* R package for modeling[95]. Kills where the wolf returned to the kill or a homesite were input as 1 whereas kills when the wolf did not return to the kill or homesite were input as 0. In both models, we evaluated the main effects of 'breeding status' of the wolf and 'ordinal day' from when the kill occurred, plus an interaction between the two variables. This approach allowed us to determine whether the probability of wolves returning to kills or homesites varied by breeding status across the pup-rearing season. Ordinal day was scaled to a mean of 0 and standard deviation of 1 prior to fitting the models for better convergence. We also included prey type (deer fawn or beaver) as a categorical variable in both models to evaluate whether the probability of a wolf returning to the kill or homesite varied by prey type. Similarly, we included prey biomass per wolf

in each model to evaluate how food abundance affected these patterns. Each model also had a random intercept term for 'wolf ID' to account for multiple kills made by the same wolf (these models were not able to support including 'year' as an additional random effect, as was done for the handling time analysis). Sex was not included in our analysis because we did not have sufficient sample size to examine sex in addition to the other covariates of interest. However, we think future analyses should investigate whether the feeding and provisioning behavior of wolves varies not only by social status but also by sex. For each model evaluated, we assessed the statistical significance of each variable based on an α value of 0.05 and whether 95% confidence intervals of parameter estimates overlapped 0. All analyses were conducted using the program R (version 4.2.2).

**Reporting summary**. Further information on research design is available in the Nature Portfolio Reporting Summary linked to this article.

## Data availability

All data and code used for this manuscript can be accessed here: https://figshare.com/s/cac83700fd18a245445c.

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

## Acknowledgements

Foremost, we thank the National Park Service/Voyageurs National Park and all park staff —including Dr. Steve Windels—involved in wolf research and monitoring efforts since 2012. Data used in this manuscript was the result, in part, of Cooperative Agreement P17AC01116 between the National Park Service and University of Minnesota. We thank the Minnesota Environment and Natural Resources Trust Fund, the University of Minnesota, Rainy Lake Conservancy, the Van Sloun Foundation, Voyageurs Conservancy, Wolf Conservation Center, International Wolf Center, The 06 Legacy, the National Wolfwatcher Coalition, Wildlife Science Center, Arc'teryx, NatureSpy, Vectronic-Aerospace, the Big Bad Project, and >6000 individual donors who have supported the work of the Voyageurs Wolf Project.

## Author contributions

All authors helped conceptualize and design the study. T.D.G. and S.J.B. led the data curation, data analysis, data visualization, and writing and revising of the manuscript.

T.D.G. and A.T.H. led the data collection efforts. A.T.H. and J.K.B. contributed substantially to revising and editing the manuscript.

## Competing interests

The authors declare no competing interests.
