## [Peer review file · Communications Biology]

nature portfolio

Peer Review File

Title : Differential provisioning roles, prey size, and prey abundance shape the dynamic feeding behavior of gray wolvesReviewers' comments:

Reviewer #1 (Remarks to the Author):

Brief summary:

The authors analysed a large well replicated (multi-year, multi pack) data set consisting of high-resolution GPS and ground-truthed kills sites. They used this to show that provisioning patterns are based on several intrinsic and extrinsic factors including breeding status and prey abundance.

Journal specific comments:

Does the manuscript have technical or conceptual flaws that should prohibit its publication? No.

Are the conclusions original? Yes.

Do you feel that the results presented are of immediate relevance for people in your own discipline or for a broader audience? Yes. The paper utilizes a huge dataset on a charismatic and ecologically important species that is hard to study. Indeed, data of this nature are hard to come by for any species. The results are robustly analysed and clearly presented, and I believe the paper will be of broad interest within and beyond the carnivore ecology discipline.

General comments:

I first want to congratulate the authors on what is a fascinating and extremely well designed, conducted, and communicated study on a substantial dataset. It was genuinely a pleasure to read. The experimental design is robust and appropriate, and the interpretation does not over-reach and seems appropriate to the results. In general, the findings are appropriately contextualised in the discussion, with the exception that perhaps the first few paragraphs were arguably too narrowly focused on the study species rather than putting it into the context of the wider literature. Subsequent paragraphs do a very good job of this however, so it is not a major criticism overall. However, I did feel that the discussion was rather long for the journal, and so I would encourage the authors to focus on the first (result heavy and wolf-centric) part of the discussion to reduce the length somewhat. Figures are all clear and valuable.

My main comment - which I hope will be alleviated by the cited manuscript being accepted in time to be cited here - is that the authors refer to a paper currently under review (Ln 194) for methods that are integral to the current submitted manuscript. If the cited manuscript is not accepted in time, the authors should provide further details here so that the reader can follow what has been done. The section that briefly details this (appropriate if the full paper accepted elsewhere) would need to be expanded if this were the case.

Second, and you allude to this in Ln 330, I also wondered how much of your pattern of feeding on fawns could be attributed to the growth of those fawns through the season? For example, what would figure 3a look like (specifically the curve for Fawn-Breeding) if fawn size were accounted for in the model?

Minor comments:

The following comments are meant only to improve flow/comprehension.

Ln 28. Is there any evidence that provisioning is related to neonate survival? I would suggest rephrasing this to end, "changes in neonate provisioning and survival".

There are a couple of paragraphs that begin with broad statements but without references. I think some citations would not go amiss in, for example, the opening sentences/statements of Ln 38 and

Ln 53. It seems amiss, for example, that a statement like, "In social carnivores, groups of breeding and non-breeding individuals help raise dependent young directly (e.g., provisioning, watching/defending dependent young) and indirectly (e.g., cooperative hunting of prey, territory maintenance)" has no references to support it.

Ln 40. Consider deleting "dependent young" from parentheses.

Ln 58. It is not necessary to cite it, but for interest Alting et al 2021 is another good example of this in African wild dogs, where dens are a long way from hunting grounds, and one or more daily recursive movement over large distances between the two sites are documented.

Ln 76/77. Consider deleting "of wolves." and ending the sentence, "their social status." instead.

I agree with the use of the term "breeders" throughout (as opposed to dominants), but I wondered whether similarly a more accurate term for subordinates would be "non-breeders". Are there no instances in the dataset of subordinate/subdominant individuals breeding? I appreciate there are species-specific standards in terminology at play here that may be adopted to ensure it sits clearly in the established literature, so again this is just a suggestion/query.

Ln 96. Consider deleting "Herein"? I think it's superfluous.

Ln 113 and more broadly. I wonder how lactation affects provisioning/handling. I would like to see a little more detail in the discussion perhaps of how the timing of weaning/lactation might affect some of the measured behaviour of breeders. I had a similar thought around Ln 268. How does the likely timing of weaning affect things?

Ln 147/148. I appreciate that you refer to another paper for more detailed methods, but I think you should also add the manufacturer details for foothold and collars here too. It would be good to make clear that they are soft-jaw traps too.

Similarly, Ln 157-159, I think the reader would benefit from knowing the delay to visit the sites rather than having to refer to a second publication. This can be stated briefly.

It would be interesting to report what the commute distances were for kills and whether this influenced their likelihood of subsequently returning to the kill.

Ln 285. For clarity and flow, consider rephrasing "to a kill than breeding" to "to a kill overall than were breeding".

Ln 307. Where you refer to Spring, can you provide the approximate months? It helps to provide context, especially for reviewers in southern/austral hemisphere.

Sentence 368-370 could be rephrased and clarified (promoting "to pups").

Ln 401 "We think" might be rephrased to "Our data support".

Reviewer #2 (Remarks to the Author):

Thank you for the opportunity to review the manuscript, "Differential provisioning roles, prey size, and prey abundance shape the dynamic feeding behavior of a social carnivore."

The foundation of this manuscript is a very impressive collection of hard-earned and rare field data used to examine the relationship between wolf foraging and the provisioning of pups, including the effects of various intrinsic and extrinsic factors. The data collected over 6 years include over 900 predation events from more than 30 GPS-collared wolves in more than 15 packs from over 17,000 clusters of GPS-locations that were searched. This vast amount of data is so challenging to amass given the cryptic nature of these low-density predators and their habitat (in this study area,

generally remote, dense forests that can be challenging to access, especially for such a comprehensive study as this). The authors acquired high-frequency GPS locations – so, not only the amount, but also the resolution of their data is what offers the opportunity to glean new insights into this area of wolf natural history. Further, the analytical approach the authors have applied is rigorous and appropriately referenced. In addition to containing critical information related to wolf natural history, this manuscript is also broadly relevant to other social predators that also utilize homesites during the rearing of young.

The manuscript is well-written and well-reasoned, and the methodological approaches are sound. The research question is well defined, relevant, and meaningful. The authors include an ethical approval statement regarding animal usage. The field methods (live-capture and radiocollaring) are common for monitoring wolves and other elusive, large predators. The authors clearly demonstrate how this research fills an identified knowledge gap. The methods are described with detail (or other publications are suitably referenced) with information sufficient for replication. The literature references and background / context provided are generally sufficient – though please see below for one suggested addition. The figures are appropriate in number and context. Underlying raw data and all R code will be made accessible. Conclusions are well stated, linked to the original research question, and appropriately limited to supporting results.

I have a few suggestions below.

The authors have cited a Demma and Mech 2009 – but there is another Demma and Mech 2009 that is more closely aligned with this topic. I suggest you include this reference as it considers wolf optimal foraging during summer.

Demma, D. J., & Mech, L. D. (2009). Wolf, *Canis lupus*, visits to white-tailed deer, *Odocoileus virginianus*, summer ranges: optimal foraging? *The Canadian Field Naturalist* 123(4):299-303.

I suggest that the authors include the phrasing “central place foragers / foraging” (e.g., Ylitalo et al., 2020) throughout the manuscript and as a key word. That phrase is an important search term in behavioral research and including it would likely result in this research being found more easily by that segment of ecologists and others outside of the “wolf world”.

I did not see an explanation in the manuscript of why sex was not included as a factor that could shape the foraging / provisioning relationship, but I noticed in the additional journal reporting that the authors submitted that it was due to sample limitations. I suggest the authors explain in the manuscript why they couldn't assess sex and suggest it as a future area of research.

Line 161 – would the handling time of prey also possibly include the time that the wolves were waiting to ambush beavers? Given other research by the Voyageurs Wolf Project on how wolves hunt beavers, whether this includes that time or not, I suggest this be more fully described so it is clear whether it likely does, does not, include waiting time.

Line 309 – re: “wolves transition from cooperative hunters that kill large ungulate prey (adult-sized deer) to solitary predators that primarily hunt and kill small prey (beavers and deer fawns)” – I believe the Voyageurs Wolf Project has evidence that wolves use cooperative strategies to ambush hunt beavers. Considering that – I think this sentence is an oversimplification. I agree there is a trend toward less pack cohesion while foraging during pup-rearing, but I am not convinced it is a transition from group to single. Perhaps a slight clarification / qualification would be appropriate.

Line 438-440 – suggestion to consider the inferential information in Demma and Mech 2009 mentioned above (not the Demma and Mech 2009 that is currently cited in the manuscript).

Authors' Response to Reviewers for COMMSBIO-23-2700
“Differential provisioning roles, prey size, and prey abundance shape the dynamic feeding behavior of a social carnivore”

Reviewers' comments:

Reviewer #1

General comments:

I first want to congratulate the authors on what is a fascinating and extremely well designed, conducted, and communicated study on a substantial dataset. It was genuinely a pleasure to read. The experimental design is robust and appropriate, and the interpretation does not over-reach and seems appropriate to the results. In general, the findings are appropriately contextualised in the discussion, with the exception that perhaps the first few paragraphs were arguably too narrowly focused on the study species rather than putting it into the context of the wider literature. Subsequent paragraphs do a very good job of this however, so it is not a major criticism overall. However, I did feel that the discussion was rather long for the journal, and so I would encourage the authors to focus on the first (result heavy and wolf-centric) part of the discussion to reduce the length somewhat. Figures are all clear and valuable.

We really appreciate the reviewer's kind words and feedback on our manuscript as well as their time spent reviewing the manuscript. We have obviously spent considerable time gathering the data and working on this manuscript, so such positive feedback was very encouraging.

We reduced the length of the first part of the discussion, per the reviewer's suggestion, by ~150 words. Much of our effort was in condensing and consolidating the section on provisioning behavior. We also changed some wording in this section to make the section more general.

My main comment - which I hope will be alleviated by the cited manuscript being accepted in time to be cited here – is that the authors refer to a paper currently under review (Ln 194) for methods that are integral to the current submitted manuscript. If the cited manuscript is not accepted in time, the authors should provide further details here so that the reader can follow what has been done. The section that briefly details this (appropriate if the full paper accepted elsewhere) would need to be expanded if this were the case.

We have removed all references to this paper because we do not think it will be through the peer-review process in time to be referenced as accepted in this article. As such, we have added additional information in our methods per the reviewer's suggestion. Specifically, we

added three detailed paragraphs describing how we estimated prey density and calculated prey biomass per wolf (L197-232).

We have also replaced any reference to this “In Review” manuscript and supported the statements with other published literature (e.g., L138 and L372).

Second, and you allude to this in Ln 330, I also wondered how much of your pattern of feeding on fawns could be attributed to the growth of those fawns through the season? For example, what would figure 3a look like (specifically the curve for Fawn-Breeding) if fawn size were accounted for in the model?

This is an excellent question and one that we considered in detail when designing our analytical approach. The problem is that it is impossible to include both Julian/ordinal day and weight of fawns during summer because any estimate of fawn weights would be directly correlated with Julian/ordinal date.

Indeed, the primary approach we are aware of to estimate fawn weights throughout the summer has been to use an equation from Rawson et al. (1992), where they examined daily growth rates of fawns from 0 to 90 days of age. By doing so they derived the following equation on the growth rate of fawns: $\log(\text{fawn body weight})=0.63+0.01*(\text{age of fawn [in days]})$.

Thus, any estimate of fawn body weight would be directly related to the age of fawns and therefore Julian date. Put differently, Julian/ordinal date is a direct proxy for fawn weight based on empirical data, and thus our modeling approach is capturing the effect of increased fawn size throughout the pup-rearing season.

And we clearly see that handling time of fawns continues to increase for breeding animals (Fig. 3a). A current challenge is that it is not possible to disentangle how much of the increase in handling time at fawn kills throughout June-August is a result of increasing fawn body size or increased maturation/mobility of pups. In reality, it is almost certainly both. We highlighted both of these possibilities in our discussion. For example, we noted that fawn handling times increased during the pup-rearing season as expected likely due to increased fawn size (L370-375) and the increase mobility and maturity of pups (L397-402).

Minor comments:

The following comments are meant only to improve flow/comprehension.

Ln 28. Is there any evidence that provisioning is related to neonate survival? I would suggest rephrasing this to end, “changes in neonate provisioning and survival”.

The reviewer suggests we change this section from “changes to neonate survival” to “changes in neonate provision and survival”. However, this change will not work because we state in the beginning of the sentence that “differential provisioning of pups” is what we

suggest is responsible for the “change in neonate survival”. It would not make sense to state that “differential provisioning of pups” is responsible for “changes in neonate provisioning”.

There are a couple of paragraphs that begin with broad statements but without references. I think some citations would not go amiss in, for example, the opening sentences/statements of Ln 38 and Ln 53. It seems amiss, for example, that a statement like, “In social carnivores, groups of breeding and non-breeding individuals help raise dependent young directly (e.g., provisioning, watching/defending dependent young) and indirectly (e.g., cooperative hunting of prey, territory maintenance)” has no references to support it.

We have added several references to support the statement in Lines 42-43. It is worth noting that in these first two introductory paragraphs, we did not include citations in the first sentence of each paragraph because the first sentence is our broad topic sentence and the rest of the paragraph expands on that topic sentence with citations to the relevant literature.

Ln 40. Consider deleting “dependent young” from parentheses.

Done!

Ln 58. It is not necessary to cite it, but for interest Alting et al 2021 is another good example of this in African wild dogs, where dens are a long way from hunting grounds, and one or more daily recursive movement over large distances between the two sites are documented.

We appreciate the reviewer mentioning this study and we have cited it in two paragraphs (L42 and L59) in our introduction because the Alting et al. 2021 paper support many of our statements in the introduction.

Ln 76/77. Consider deleting “of wolves.” and ending the sentence, “their social status.” instead.

Done! Good suggestion!

I agree with the use of the term "breeders" throughout (as opposed to dominants), but I wondered whether similarly a more accurate term for subordinates would be “non-breeders”. Are there no instances in the dataset of subordinate/subdominant individuals breeding? I appreciate there are species-specific standards in terminology at play here that may be adopted to ensure it sits clearly in the established literature, so again this is just a suggestion/query.

The reviewer raises a good point regarding terminology here. We used the terms breeder and subordinates in our manuscript based on the highly-influential paper by Mech (1999. Alpha status, dominance, and division of labor in wolf packs. Canadian Journal of Zoology), in which he discussed in detail the typical structure of wolf packs and the role of dominance in packs. Mech’s conclusion was that packs largely consist of a breeding pair and their subordinate offspring which he refers to as breeders and subordinates. Most of the time subordinates are the direct offspring of the breeding pair but at times,

subordinates might be unrelated to the breeding pair or a sibling of the breeding pair. However, as far as we are aware there were no subordinate wolves that were breeders in these data, and such an occurrence is exceptionally rare in areas such as northern Minnesota where pack sizes are small (4-5 wolves) and pack structure very simple (largely parents and their offspring).

Ln 96. Consider deleting “Herein”? I think it’s superfluous.

Done!

Ln 113 and more broadly. I wonder how lactation affects provisioning/handling. I would like to see a little more detail in the discussion perhaps of how the timing of weaning/lactation might affect some of the measured behaviour of breeders. I had a similar thought around Ln 268. How does the likely timing of weaning affect things?

Wolf pups are weaned relatively quickly and by about 4-5 weeks of age (1st-2nd week in May), pups are eating solid food. Thus, the period when wolf pups are provisioned primarily via lactation constitutes only about ~20% of the pup-rearing season (April to August). In the weeks following parturition, breeding females remain largely with the pups and breeding males provision breeding females with food (Mech 1999).

There is little in our data that indicates that weaning of pups has a substantial impact on any of the feeding or provisioning metrics that we observed. In Figure 2, for example, we see no trend to indicate an abrupt change in handling time before or after pups are weaned, though there is a slight, gradual increase in handling time from April to June. However, we think this increase is due to increased maturity of pups and their ability to go longer periods without food as opposed to a change provisioning behavior (i.e., regurgitation instead of lactation).

To address the reviewer’s comment here, we have added a statement to the discussion noting that pups are provisioned by a lactating female who is largely provisioned by the breeding male for 4-5 weeks after parturition (L390-392). However, we are not able to provide any detailed discussion about how lactation influences handling time as it is not clear to us if or to what extent lactation does or does not influence the trends we have identified. For example, even once pups are weaned, breeding wolves spend very brief periods of time at kills relative to subordinate wolves.

Ln 147/148. I appreciate that you refer to another paper for more detailed methods, but I think you should also add the manufacturer details for foothold and collars here too. It would be good to make clear that they are soft-jaw traps too.

Done! L148-150

Similarly, Ln 157-159, I think the reader would benefit from knowing the delay to visit the sites rather than having to refer to a second publication. This can be stated briefly.

Done! L161

It would be interesting to report what the commute distances were for kills and whether this influenced their likelihood of subsequently returning to the kill.

We agree this would be a very interesting analysis and is something we are planning to examine in the future in our attempt to understand the foraging behavior of wolves during summer. However, including this information and analysis would make this paper substantially longer and we think it best to reserve this analysis and assessment for a separate manuscript where the topic can be examined in greater and adequate detail.

Ln 285. For clarity and flow, consider rephrasing “to a kill than breeding” to “to a kill overall than were breeding”.

Done! We have rephrased this. L328.

Ln 307. Where you refer to Spring, can you provide the approximate months? It helps to provide context, especially for reviewers in southern/austral hemisphere.

Good suggestion. We removed one reference to spring and replaced with “early April” to be more specific. In the other use of spring (L72), we added the months we were considering to be spring.

Sentence 368-370 could be rephrased and clarified (promoting “to pups”).

We are not really sure what the reviewer is referring to with ‘promoting to pups’. We re-read this sentence several times and are not sure what needs to be clarified or was unclear so we left it as is.

In 401 “We think” might be rephrased to “Our data support”.

This sentence has been changed in our effort to condense our discussion and this comment not relevant given those changes.

Reviewer #2:

Thank you for the opportunity to review the manuscript, “Differential provisioning roles, prey size, and prey abundance shape the dynamic feeding behavior of a social carnivore.”

The foundation of this manuscript is a very impressive collection of hard-earned and rare field data used to examine the relationship between wolf foraging and the provisioning of pups, including the effects of various intrinsic and extrinsic factors. The data collected over 6 years include over 900 predation events from more than 30 GPS-collared wolves in more than 15 packs from over 17,000 clusters of GPS-locations that were searched. This vast amount of data is so challenging to amass given the cryptic nature of these low-density predators and their habitat (in this study area, generally remote, dense forests that can be challenging to access, especially

for such a comprehensive study as this). The authors acquired high-frequency GPS locations – so, not only the amount, but also the resolution of their data is what offers the opportunity to glean new insights into this area of wolf natural history. Further, the analytical approach the authors have applied is rigorous and appropriately referenced. In addition to containing critical information related to wolf natural history, this manuscript is also broadly relevant to other social predators that also utilize homesites during the rearing of young.

The manuscript is well-written and well-reasoned, and the methodological approaches are sound. The research question is well defined, relevant, and meaningful. The authors include an ethical approval statement regarding animal usage. The field methods (live-capture and radiocollaring) are common for monitoring wolves and other elusive, large predators. The authors clearly demonstrate how this research fills an identified knowledge gap. The methods are described with detail (or other publications are suitably referenced) with information sufficient for replication. The literature references and background / context provided are generally sufficient – though please see below for one suggested addition. The figures are appropriate in number and context. Underlying raw data and all R code will be made accessible. Conclusions are well stated, linked to the original research question, and appropriately limited to supporting results.

We sincerely appreciate the positive and encouraging feedback from the reviewer. We are pleased they found our approach and analysis rigorous and our conclusions well-justified and reasoned.

I have a few suggestions below.

The authors have cited a Demma and Mech 2009 – but there is another Demma and Mech 2009 that is more closely aligned with this topic. I suggest you include this reference as it considers wolf optimal foraging during summer.

Demma, D. J., & Mech, L. D. (2009). Wolf, *Canis lupus*, visits to white-tailed deer, *Odocoileus virginianus*, summer ranges: optimal foraging? *The Canadian Field Naturalist* 123(4):299-303.

Done! We have added a reference to the Demma and Mech 2009 article in Line 478-479.

I suggest that the authors include the phrasing “central place foragers / foraging” (e.g., Ylitalo et al., 2020) throughout the manuscript and as a key word. That phrase is an important search term in behavioral research and including it would likely result in this research being found more easily by that segment of ecologists and others outside of the “wolf world”.

We have included the phrase “central place foraging” in the keywords and added some additional references to “foraging” in the manuscript. The term foraging is now referenced 16 times in the manuscript.

Notably, foraging is defined as “to wander in search of forage or food” (Merriam-Webster Dictionary, and to “search widely for food or provisions” (Oxford Languages). The behaviors we examined throughout the manuscript were those that occurred once wolves had already procured food, and thus we think “feeding behavior” more accurately

describes what we studied than the term “foraging behavior”.

I did not see an explanation in the manuscript of why sex was not included as a factor that could shape the foraging / provisioning relationship, but I noticed in the additional journal reporting that the authors submitted that it was due to sample limitations. I suggest the authors explain in the manuscript why they couldn't assess sex and suggest it as a future area of research.

Done! L279-282.

Line 161 – would the handling time of prey also possibly include the time that the wolves were waiting to ambush beavers? Given other research by the Voyageurs Wolf Project on how wolves hunt beavers, whether this includes that time or not, I suggest this be more fully described so it is clear whether it likely does, does not, include waiting time.

In instances where wolves killed beavers via sit-and-wait ambush strategies, handling time would include the time spent waiting in ambush because we are not able to determine how long a wolf waited in ambush before killing a beaver using GPS-collar data (see Gable et al. 2021 for a detailed discussion of why). If we were able to determine this, we could ascertain when a beaver was killed and subtract out the time a wolf spent ambushing a beaver. Unfortunately, this is not currently possible with our data collection methods.

Worth noting: while ambushing may influence some handling time estimates, we suspect the influence on our results is minimal and would have little impact on our findings. We have data that indicates that for most of the pup-rearing season (April to August), wolves are not hunting beavers via ambush strategies (*in prep*). Instead, wolf ambush rates appear to peak in late August and then slowly decline through September and October—a pattern we postulate is driven by reduced prey availability and increased energetic demands of wolf pups in late summer and early fall. Regardless, for most of the pup rearing season wolves are primarily hunting and killing beavers largely via opportunistic or cursorial strategies.

In addition, we have data to indicate that there are wolves that ambush beavers and wolves that do not (Bump et al 2022). Indeed, a sizable proportion of the wolves we have studied have spent very little time waiting in ambush for beavers. We are still trying to determine why we see such variability in the hunting strategies of wolves in our area but hope further work will shed light on this.

Nonetheless, we added a few sentences to our methods section to clarify that handling time may include the time wolves spent waiting-in-ambush for beavers per the reviewer's suggestions (L173-178).

Line 309 – re: “wolves transition from cooperative hunters that kill large ungulate prey (adult-sized deer) to solitary predators that primarily hunt and kill small prey (beavers and deer fawns)” – I believe the Voyageurs Wolf Project has evidence that wolves use cooperative strategies to ambush hunt beavers. Considering that – I think this sentence is an oversimplification. I agree

there is a trend toward less pack cohesion while foraging during pup-rearing, but I am not convinced it is a transition from group to single. Perhaps a slight clarification / qualification would be appropriate.

We have clarified that this is referring to wolves in our study system and not necessarily wolves in other systems (L351). We have also tweaked the wording to indicate that wolves are “largely solitary predators” during this time (L352). As the data we presented in the manuscript indicates, wolves were by themselves (i.e., not with another collared wolf) at 94% of the kills during April to August (L295), which we think is strong evidence that wolves are largely solitary predators during this period.

Line 438-440 – suggestion to consider the inferential information in Demma and Mech 2009 mentioned above (not the Demma and Mech 2009 that is currently cited in the manuscript).

We re-read the Demma and Mech 2009 paper that the reviewer has referenced here. We did cite this paper elsewhere in our manuscript (see above) but after reading it, we are unsure how this paper supports any of the statements mentioned here. As such, we have decided not to cite it to support these statements.

REVIEWERS' COMMENTS:

Reviewer #1 (Remarks to the Author):

The authors have done a fantastic job in addressing all comments and suggestions from the previous review. Where they have not implemented changes, the authors have provided adequate and well-reasoned arguments for not doing so and edited the text accordingly to reflect this. I have no further suggestions, and think that the paper is of a very high quality and will be of broad interest to the readers of the journal.

Reviewer #2 (Remarks to the Author):

Hello, the authors have done a great job suitably justifying the changes they made (or changes they elected not to make) in response to reviewer comments. I have no further suggestions. Thank you.